# EXPLORING REPRENSENTATION INVARIANCE IN FINE-TUNING

## ABSTRACT

Foundation models pretrained on large-scale natural images are widely adapted to various cross-domain low-resource downstream tasks, benefiting from generalizable and transferable patterns captured by their representations. However, these representations are later found to gradually vanish during finetuning, accompanied by a degradation of model's original generalizability. In this paper, we argue that such tasks can be effectively adapted without sacrificing the benefits of pretrained representations. We approach this by introducing *Representation Invariance FineTuning (RIFT)*, a regularization that maximizes the representation similarity between pretrained and finetuned models by leveraging orthogonal invariance of manifolds in a computationally efficient way. Experiments demonstrate that our method is compatible with mainstream finetuning methods, offering competitive or even enhanced performance and better preservation of the generalizability.

## 1 INTRODUCTION

Foundation models pretrained on large-scale natural images have been widely recognized as strong initializations for various downstream tasks Siméoni et al. (2025); Yu et al. (2025); Fang et al. (2023), particularly in data-scarce scenarios Liu et al. (2024a); Zhang et al. (2024a;b), as low-resource learning tasks usually fail to train a powerful model to cover the complex data distributions, resulting in poor performance Tian et al. (2020). A representative example is medical image analysis Wang et al. (2023); Dai et al. (2023), where collecting labeled samples is difficult due to privacy concerns, rare diseases, heterogeneous sources, and the high annotation cost. In contrast, the generalizable and transferable patterns captured by pretrained representations can strongly compensate for this limitation, which are shown to be shared across tasks Mehra et al. (2024).

Meanwhile, such tasks typically exhibit significant domain gaps, which necessitate finetuning (*i.e.*, training on specific datasets with smaller learning rates, fewer epochs and selective parameters) to successfully adapt foundation models Jia et al. (2022b); Chen et al. (2022); Hu et al. (2022). The primary intention of finetuning is to implicitly minimize Euclidean distance shift of pretrained model in parameter space, thereby not only enabling better transfer of established semantic knowledge to downstream tasks but also preserving the inherent generalizability to support future incremental requirements, such as continual and multi-task learning Wang et al. (2024); Zhang & Yang (2021).

However, these benefits of pretrained representations are not retained as expected, and instead degrade severely during finetuning Wang et al. (2025); Chen et al. (2025); Kumar et al. (2022). Particularly, recent Platonic Representation Hypothesis reveals that foundation models possess enough capacity and scalability to capture non-conflicting shared representations across modalities and tasks, yet this potential is often wasted, as simplicity bias may converge along shortcut paths tailored to the current task Huh et al. (2024). This leaves us wondering, *how can pretrained representations avoid being shaved away by such an Occam's razor in cross-domain low-resource finetuning?*

Many biological systems adjust to new environments while protecting their core functions, a stability that arises from constraints imposed throughout the evolution, *e.g.*, protein structures influenced by hydrophobic interactions remain robust in their overall fold even when large temperature fluctuations perturb local residues Hatakeyama & Kaneko (2015); Tang et al. (2020). Inspired by this principle, we hypothesize that pretrained and finetuned representations fail to coexist due to insufficient constraints.

Intuitively, one can restrict finetuning by aligning it with pretrained representations to prevent excessive drift. This alignment can be achieved through similarity measures, with Centered Kernel Alignment (CKA) as a popular choice Kornblith et al. (2019). A straightforward approach is to add a regularization term that keeps representation similarity above certain threshold to the loss function, but it incurs considerable computational cost due to the high complexity of pairwise CKA calculations. To alleviate this, we propose *Representaion Invariance FineTuning (RIFT)* to make three efforts. First, we exploit the orthogonal invariance of CKA by maintaining orthogonality between two feature embeddings, as a cheaper alternative. Second, we perform distributional rather than sample-wise alignment for each mini-batch, leveraging the precomputed mean and covariance for an efficient statistical approximation. Third, we use only feature embeddings from the last layer before task head to avoid orthogonality violations in multi-layer architectures or nonlinear activations.

Overall, our contributions can be summarized as follows:

- We demonstrate that the benefits of pretrained representations can be well preserved while still effectively adapting to downstream tasks, especially in cross-domain low-resource scenarios.
- We propose **RIFT**, a simple regularization for finetuning to constrain representation similarity by the orthogonal invariance of CKA with improved computational efficiency.
- Our method is compatible with mainstream finetuning methods, achieving competitive or even enhanced performance while better preserve the generalizability.

We hope our work can shed some light on the finetuning paradigms that emphasize both generalization and adaptation.

## 2 RELATED WORK

### 2.1 REPRESENTATION INVARIANCE AND SIMILARITY

Pretrained representations have been shown to capture rich and diverse features from real-world datasets, and are widely used to accelerate and stabilize the convergence of downstream tasks Yu et al. (2025); Wu et al. (2023); He et al. (2022); Liang et al. (2025). However, these representations have been found to inevitably degrade during finetuning, with downstream task performance being adversely affected and catastrophic forgetting also weakening pretrained semantic knowledge. Consequently, several studies have proposed methods to address these issues Aghajanyan et al. (2021); Razdaibiedina et al. (2023); Ma et al. (2021). Yet, despite these efforts, the generalizability of pretrained representation is neglected, and remains unverified as inevitable degradation, with no effective approaches yet established Wang et al. (2025); Chen et al. (2025); Kumar et al. (2022). In this paper, we confront the challenge of whether pretrained representations and their generalizability can be simultaneously preserved, *i.e.*, exploring the representation invariance in finetuning.

Representation similarity metrics provide a natural tool to quantify such invariance. Different similarity measures Klabunde et al. (2025); Huh et al. (2024) have been proposed to compare representations across layers or models to better understand neural network behaviors, *e.g.*, Canonical Correlation Analysis (CCA) Morcos et al. (2018) and Centered Kernel Alignment (CKA) Kornblith et al. (2019). These metrics exhibit various desirable invariances, such as invariance to invertible linear transformations, orthogonal transformations, and isotropic scaling. Among them, CKA offers superior consistency across architectures, stronger invariance, and more interpretable results.

### 2.2 FINETUNING AND ORTHOGONAL CONSTRAINTS

Typically, finetuning is divided into full finetuning Lv et al. (2024) and parameter-efficient finetuning (PEFT) Ding et al. (2023). Full finetuning retrains all pretrained parameters Kirillov et al. (2023); Touvron et al. (2023); Liu et al. (2023), while PEFT updates only a subset or newly introduced ones, *e.g.*, prompt tuning Zu et al. (2024); Jia et al. (2022b); Bahng et al. (2022), adapter tuning Chen et al. (2022); He et al. (2021); Sung et al. (2022), and LoRA Hu et al. (2022); Liu et al. (2024b); Hayou et al. (2024).

Introducing orthogonal constraints during training and finetuning Qiu et al. (2023); Liu et al. (2024c); Ma et al. (2024); Yang et al. (2025); Qiu et al. (2025); Duan et al. (2025) has been proposed to

preserve pretrained generative abilities. The motivation lies in the fact that orthogonal transformations preserve both the spectral norm of weight matrices and angular relationships Qiu et al. (2023; 2025). In particular, OFT Qiu et al. (2023) enforces such constraints on the weight matrix to maintain hyperspherical energy. *In contrast, RIFT is fundamentally different from these approaches: (1) RIFT operates directly on the final representations rather than imposing constraints on specific weights, thereby avoiding the limitations detailed in Sec. 4.3. (2) RIFT prioritizes preserving the generalizability of pretrained representations, rather than solely transferring established semantic knowledge to downstream tasks, though it achieves this as well.*

# 3 PRELIMINARY: CKA FOR REPRESENTATION SIMILARITY

**Notation.** Bold symbols denote matrices or vectors (e.g., $\mathbf{X}$ and $\mathbf{x}$), while scalars are written in lowercase (e.g., $x$). The dataset $\mathbf{X} = \{(\mathbf{x}_i)\}_{i=1}^n \in \mathbb{R}^{n \times d}$ where $n$ is the number of samples and $d$ is feature dimension. The Frobenius norm is denoted by $\| \cdot \|_F$.

**Model.** A model is represented by $f_\theta$, where $\theta$ denotes the parameters, and we simplify it as $f$. The notation $\mathbf{F}_\theta(\mathbf{X})$ denotes features extracted by backbone, which is distinct from the classification output $f_\theta(\mathbf{X})$. They are omitted as $\mathbf{X}$ and $\mathbf{F}_\theta$. The parameters of pretrained model are denoted as $\theta_0$.

CKA has been a widely used metric for representation similarity. Formally, the centered feature embedding $\mathbf{F}_{\theta,c} \in \mathbb{R}^{n \times d}$ computed by $f_\theta$ on $\mathbf{X}$ is defined as

$$\mathbf{F}_{\theta,c} := \mathbf{F}_\theta - \frac{1}{n}\mathbf{1}\mathbf{1}^\top \mathbf{F}_\theta \tag{1}$$

where $\mathbf{1} \in \mathbb{R}^n$ is the all-ones vector. The linear CKA of $\mathbf{F}$ between $f_{\theta_0}$ and $f_\theta$ is then given by

$$\mathrm{CKA}(\mathbf{F}_{\theta_0}, \mathbf{F}_\theta) = \frac{\left\|\mathbf{F}_{\theta_0,c}^\top \mathbf{F}_{\theta,c}\right\|_F^2}{\left\|\mathbf{F}_{\theta,c}^\top \mathbf{F}_{\theta,c}\right\|_F \left\|\mathbf{F}_{\theta_0,c}^\top \mathbf{F}_{\theta_0,c}\right\|_F} \tag{2}$$

which is the cosine similarity of centered features' Gram matrices and quantifies the structural alignment degree. It is particularly sensitive to dimensional collapse, while remaining invariant to orthogonal transformations and isotropic scalings. The invariance is stated as

**Property 3.1** (CKA Similarity-Transformation Invariance). *For any scalar $\alpha > 0$ and orthogonal matrix $\mathbf{Q} \in O(d)$, $\mathbf{F}_\theta = \alpha \mathbf{F}_{\theta_0}\mathbf{Q}$ satisfies*

$$CKA(\mathbf{F}_{\theta_0}, \mathbf{F}_\theta) = 1 \tag{3}$$

where such transformations ensure a high CKA value, implying that feature embeddings remain highly similar. This condition motivates the following definition.

**Definition 3.2** (Similarity-Invariant Parameter Subspace). *Given $\theta_0$, define the set*

$$\mathcal{M}_{\theta_0} := \{\theta \in \Theta \mid \exists \mathbf{Q} \in O(d), \exists \alpha > 0 \ such \ that \ \mathbf{F}_\theta = \alpha \mathbf{F}_{\theta_0}\mathbf{Q}\}. \tag{4}$$

which offers a promising way to identify finetuned models preserving the pretrained representation, and we primarily focus on feature embeddings from the last layer before task head.

# 4 METHOD

## 4.1 PROBLEM FORMULATION

The goal is to find a finetuned model that both maximizes downstream task performance and preserves representation similarity with pretrained model. This is formulated as an optimization problem:

$$\theta^* = \arg\min_\theta \mathcal{L}_{\mathrm{cls}}(\theta) \quad \text{s.t.} \quad \mathrm{CKA}(\mathbf{F}_{\theta_0}, \mathbf{F}_\theta) \geq \epsilon \tag{5}$$

where $\mathcal{L}_{\mathrm{cls}}(\theta)$ is the classification loss and $\epsilon \in [0, 1]$ is threshold. To solve Eq. 5, we can introduce the *Representation Similarity Constrained (RSC)* Loss and try to minimize it:

$$\mathcal{L}_{\mathrm{RSC}}(\theta) = \mathcal{L}_{\mathrm{cls}}(\theta) + \lambda \mathcal{L}_{\mathrm{CKA}}(\theta) \tag{6}$$

$$\mathcal{L}_{\mathrm{CKA}}(\theta) = 1 - \mathrm{CKA}(\mathbf{F}_{\theta_0}, \mathbf{F}_\theta) \tag{7}$$

where $\mathcal{L}_{\mathrm{CKA}}(\theta)$ is the similarity loss, and $\lambda$ is the regularization strength. However, computing $\mathcal{L}_{\mathrm{CKA}}(\theta)$ causes considerable computational overhead, as pairwise CKA incurs high complexity.

## 4.2 REPRESENTATION INVARIANCE FINETUNING

To efficiently preserve representation similarity, we reformulate Eq. 5 with Definition 3.2 as

$$\theta^* = \arg\min_{\theta \in \mathcal{M}_{\theta_0}} \mathcal{L}(\theta) \tag{8}$$

which implicitly guarantees representation similarity, and replaces the explicit CKA computation. Given a learnable orthogonal matrix $\mathbf{Q} \in \mathbb{R}^{d \times d}$ and scaling factor $\alpha$, we enforce the finetuned representation to reside on orthogonal manifold of pretrained representation with Property 3.1 by

$$\left\| \mathbf{F}_\theta(\mathbf{X}) - \alpha \mathbf{F}_{\theta_0}(\mathbf{X})\mathbf{Q} \right\|_F^2 \tag{9}$$

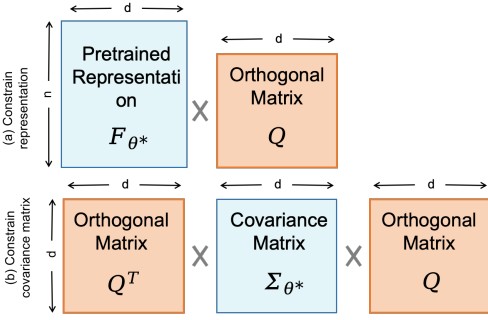

Figure 1: (a) Orthogonal transformation of the pretrained representation. (b) Orthogonal transformation of the covariance.

Notably, directly imposing this constraint either demands a sample-wise forward pass through the pretrained model or substantial storage. We instead approximate it by matching the mini-batch $\boldsymbol{\mu} \in \mathbb{R}^d$ and covariance $\boldsymbol{\Sigma} \in \mathbb{R}^{d \times d}$ of for every mini-batch:

$$\left\| \boldsymbol{\mu}_\theta - \alpha \boldsymbol{\mu}_{\theta_0}\mathbf{Q} \right\|_F^2 \quad \text{and} \quad \left\| \boldsymbol{\Sigma}_\theta - \alpha^2 \mathbf{Q}^\top \boldsymbol{\Sigma}_{\theta_0}\mathbf{Q} \right\|_F^2 \tag{10}$$

The analysis of this relaxation rationality is provided in Theorem A3.5. In practice, we find that the transformation $\alpha\mathbf{Q}$ has already adjusted both scale and orientation of pretrained representations, implicitly aligning their mean with current representations. We also observe that dynamic $\alpha$ brings little convergence benefits and increases training burden. Therefore, we simplify Eq. 10 to covariance matching with $\alpha = 1$, and further propose the *Representation Invariance FineTuning (RIFT)* Loss:

$$\mathcal{L}_{\text{RIFT}}(\theta, \mathbf{Q}) = \mathcal{L}_{\text{cls}}(\theta) + \mathcal{L}_{\text{Cov}}(\theta, \mathbf{Q}) \tag{11}$$

$$\mathcal{L}_{\text{Cov}}(\theta, \mathbf{Q}) = \left\| \boldsymbol{\Sigma}_\theta - \mathbf{Q}^\top \boldsymbol{\Sigma}_{\theta_0}\mathbf{Q} \right\|_F^2 \tag{12}$$

**Generalization and Adaptation.** Here, $\mathbf{Q}$ applies only global isometric rotations to the whole feature space without altering relative angles among different class clusters or sample vectors, thereby preserving semantic consistency and generalizability of the pretrained representation, *i.e.*, any sample, whether in- or out-of-pretrained-distribution, are mapped to its original predicted label after the current orthogonal transformation. With this foundation, $\mathcal{L}_{\text{cls}}(\theta)$ promotes the representation to generate additional semantic structures for finetuning datasets.

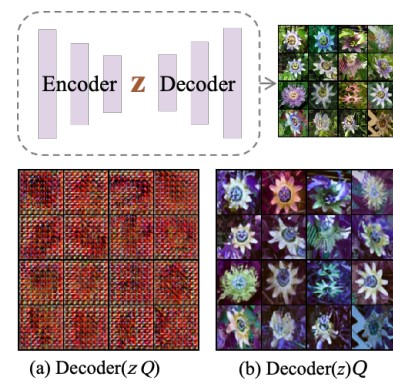

(a) Decoder($z\,Q$)   (b) Decoder($z$)$Q$

Figure 2: (a) Applying Orthogonal transformation at the intermediate layer. (b) Applying orthogonal transformation at the last layer.

**Compatibility and Efficiency.** Importantly, RIFT reveals that generalization and adaptation are not mutually exclusive, and can serve as a plug-and-play training strategy compatible with mainstream finetuning methods, since it constrains only the output representations without modifying networks. RIFT can also be easily integrated with more advanced orthogonal finetuning techniques by updating $\mathbf{Q}$. Through trace-based decoupling of covariance from individual samples, the computational complexity of RIFT is reduced from $O(nd^2)$ to $O(d^2)$. Detailed training time is presented in Tab. A5.

## 4.3 WHY ORTHOGONAL CONSTRAINTS IMPOSED ONLY AT FINAL FEATURE EMBEDDINGS

An intriguing question is why we apply the orthogonal transformation to last-layer feature embeddings before task head rather than output of another or each layer. The reason initially lies in fact that final feature embeddings are used for task decisions, which must be kept. Although imposing individual constraints on a single linear layer guarantees similar outputs, extending it to multi-layer models or introducing nonlinearities (*e.g.*, activations) breaks the full model representation invariance.

We first expand the network depth and apply orthogonal transformations on just one layer, using an autoencoder trained for image reconstruction in Fig 2. As shown in Fig. 2 (b), applying random

orthogonal transformations to the final feature embeddings of pretrained model allows for clear image reconstruction. In contrast, results in Fig 2 (a) fail because constraint is only added to the encoder, while the decoder remains untransformed, causing a distribution mismatch and thus hindering the complete recovery of feature embeddings in deeper layers. This demonstrates that applying constraints at other layers cannot guarantee the invariance of final feature embeddings.

Then we further apply layer-wise random independent orthogonal transformations, and conduct a toy experiment in Fig. 3 with two Gaussian-distributed classes on 6-layer linear or nonlinear networks. The results indicate that they still results in severe representation degradation in deep networks with nonlinearities further making the invariance preservation uncertain. These findings highlight constraining final feature embeddings is the key to global representation invariance.

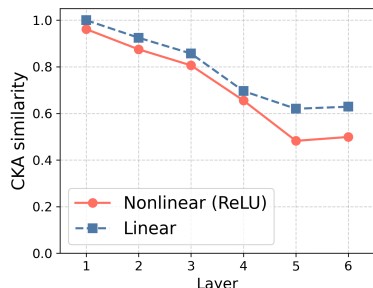

Figure 3: Representation similarity diminishes with layer-wise orthogonal constraints.

## 5 EVALUATION

### 5.1 SETTING

We choose medical image classification as the representative cross-domain low-resource downstream task. The backbone is a Vision Transformer Dosovitskiy et al. (2020) (ViT-B/16), pretrained on ImageNet-1K (12 encoder layers, 768-dim embeddings, 12 attention heads). We compare our method RIFT against RSC by integrating them into mainstream finetuning methods, including full finetuning (FULL), classification-head only (LINEAR), training from scratch (SCRATCH), and PEFT(VPT, AdaptFormer, LoRA). Evaluation metrics contains CKA for representation similarity measure (Sim) as well as accuracy (Acc) and mean Average Precision (mAP) for classification performance. For reference, the representaion similarity of LINEAR is always 1 since backbone is frozen. Experiments are conducted on five medical datasets, including three MedFM subsets Wang et al. (2023) (Chest, Colon, Endo), ISIC2018 Codella et al. (2019), and APTOS2019 Karthik & Dane. They covers diverse modalities (skin, fundus, chest X-ray, pathology, colonoscopy) and tasks (multi-class, binary, multi-label). Additional training and evaluation details are provided in Appendix A1.

### 5.2 MAIN RESULT

#### 5.2.1 FINETUNING RESULT

**Quantitative result on medical image classification.** The results in Table 1 demonstrate that RIFT consistently improves representation similarity across both single-label and multi-label datasets, while maintaining competitive and even enhanced accuracy. For non-PEFT methods, although FULL finetuning achieves higher Acc than SCRATCH, it often results in lower Sim, highlighting the degradation of pretrained representation with FULL. In contrast, both RIFT and RIFT* are able to preserve or even enhance Sim compared to FULL, with RIFT* achieving the best balance between downstream task performance and pretrained representation preservation across datasets. Notably, on single-label datasets such as ISIC2018 and APTOS2019, RIFT variants maintain Acc close to FULL, while substantially improving Sim by 50–87.5%, indicating that the learned patterns remain closer to pretrained feature space, which is critical for transferability and generalization. Compared to RSC, RIFT achieves higher representation similarity with better downstream adaptation, suggesting that orthogonal transformations provide a more effective constraint than direct CKA-based regularization.

For PEFT methods, including LoRA, Adaptformer, and VPT, the trends are similar but more pronounced. While these methods already restrict parameter updates to preserve pretrained knowledge, applying RIFT still yields substantial gains in Sim, often exceeding 15–28%, while the Acc remains comparable or slightly higher in RIFT* variants. This demonstrates that RIFT effectively regularizes the adaptation process without sacrificing performance, leading to representations both task-effective and semantically faithful to the original model. Moreover, across multi-label datasets, where maintaining correlations between multiple outputs is crucial, RIFT significantly enhances Sim, suggesting

Table 1: **Quantitative result on medical image classification.** For each task, the average result from 3 runs is reported. Dataset type is explicitly indicated: single-label (left) vs. multi-label (right). Methods are grouped into non-PEFT and PEFT for clarity. **Bold** indicates the best performance. Values in the last column show relative changes (%): +/- denote increase/decrease relative to the first row of each block. RIFT refers to the default setting with regularization coefficient $\lambda = 1$, while RIFT* and RSC* correspond to the best $\lambda$ selected for each dataset.

| Model | Single-label datasets | | | | | | Multi-label datasets | | | | Average | |
|---|---|---|---|---|---|---|---|---|---|---|---|---|
| | ISIC2018 (7) | | APTOS2019 (5) | | MedFM-Colon (2) | | MedFM-Chest (19) | | MedFM-Endo (4) | | | |
| | Acc | Sim | Acc | Sim | Acc | Sim | mAP | Sim | mAP | Sim | Acc/mAP | Sim |
| **Non-PEFT Methods** | | | | | | | | | | | | |
| SCRATCH | 66.91 | 0.36 | 69.49 | 0.37 | 89.42 | 0.67 | 11.89 | 0.47 | 17.08 | 0.39 | 50.96 | 0.45 |
| FULL | 84.63 | 0.32 | 84.06 | 0.52 | **99.74** | 0.58 | 33.56 | 0.58 | 58.11 | 0.61 | 72.02 | 0.52 |
| +RSC* | **85.19** | 0.35 | 84.97 | 0.59 | 99.51 | 0.82 | 34.62 | 0.44 | 57.52 | 0.67 | +0.47% | +9.62% |
| **+RIFT (ours)** | 83.95 | **0.60** | 83.61 | **0.73** | 99.05 | 0.82 | 33.11 | 0.55 | 55.42 | **0.70** | -1.37% | +30.77% |
| **+RIFT* (ours)** | 84.85 | 0.48 | **85.25** | 0.54 | 99.51 | **0.84** | **35.24** | **0.63** | **58.62** | 0.66 | +0.93% | +21.15% |
| **PEFT Methods** | | | | | | | | | | | | |
| LINEAR | 73.57 | 1.00 | 77.87 | 1.00 | 94.24 | 1.00 | 23.55 | 1.00 | 37.80 | 1.00 | 61.41 | 1.00 |
| LoRA Hu et al. (2022) | 80.85 | 0.38 | 81.01 | 0.61 | 96.17 | 0.78 | **25.91** | 0.59 | 39.56 | 0.77 | 64.70 | 0.63 |
| **+RIFT(ours)** | 77.51 | **0.74** | 79.24 | **0.79** | 96.07 | **0.82** | 24.28 | **0.63** | 39.24 | 0.76 | -2.22% | +20.18% |
| **+RIFT*(ours)** | **80.92** | 0.40 | **81.01** | 0.61 | **96.37** | 0.79 | 25.77 | 0.61 | **39.88** | **0.77** | +0.14% | +1.59% |
| Adaptformer Chen et al. (2022) | **80.42** | 0.51 | 80.60 | 0.75 | 97.25 | 0.78 | 25.13 | 0.61 | 40.08 | 0.67 | 64.69 | 0.66 |
| **+RIFT(ours)** | 77.65 | **0.74** | 80.15 | **0.85** | 95.87 | **0.86** | 23.82 | 0.59 | 38.63 | **0.75** | -2.27% | +15.15% |
| **+RIFT*(ours)** | 79.96 | 0.56 | **80.60** | 0.81 | **97.54** | 0.81 | **25.24** | **0.65** | **40.55** | 0.71 | +0.14% | +7.58% |
| VPT Jia et al. (2022a) | 77.71 | 0.46 | 78.96 | 0.60 | 94.79 | 0.72 | 24.00 | 0.51 | **41.64** | 0.65 | 63.42 | 0.59 |
| **+RIFT(ours)** | 76.32 | **0.74** | 78.69 | **0.85** | 95.48 | 0.86 | 22.19 | **0.59** | 37.70 | **0.75** | -2.11% | +28.81% |
| **+RIFT*(ours)** | **78.31** | 0.50 | **78.96** | 0.66 | **95.48** | **0.86** | 24.58 | 0.47 | 40.54 | 0.67 | +0.24% | +4.43% |

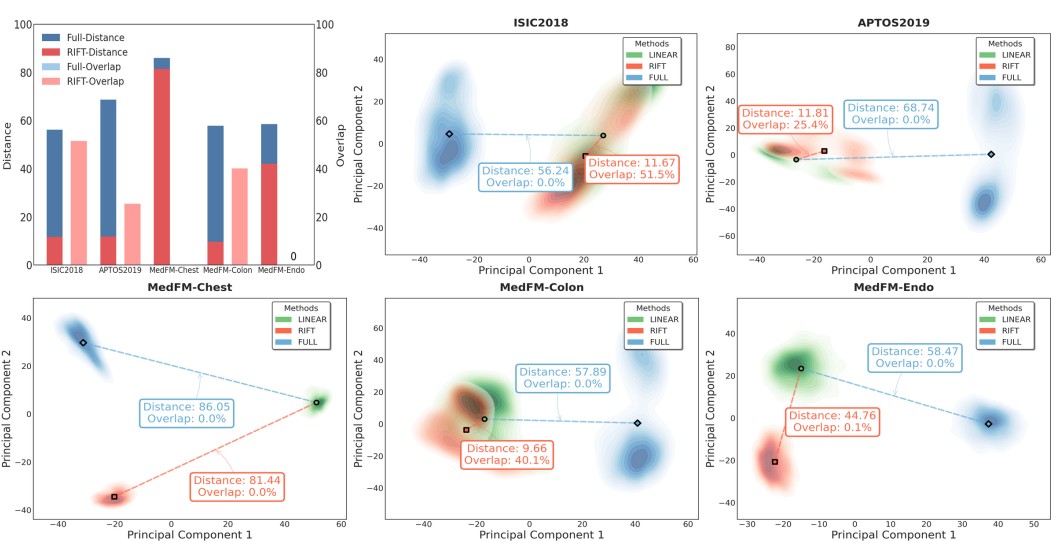

Figure 4: **PCA distribution visualization.** We compare first two principal components of LINEAR, FULL, and FULL+RIFT($\lambda = 1$) features across five medical image datasets. The first figure summarizes the distance and overlap of the pretrained model features. Darker colors indicate higher feature density, while lighter colors indicate lower density.

better preservation of semantic structure. By coupling adaptation with representation similarity, RIFT preserves established semantic knowledge, allowing it to serve downstream tasks more effectively.

**PCA distribution visualization.** Fig. 4 shows the PCA feature distribution results of samples extracted from pretrained and finetuned models. The bar charts summarize the distance from the pretrained model's feature center to the finetuned model's feature center, as well as the overlap between features. FULL shows a noticeable distance and lack of overlap between features, indicating significant representation differences compared to LINEAR. In contrast, RIFT exhibits a high overlap of pretrained features on the ISIC2018, APTOS2019, and MedFM-Colon datasets. Even

for the multi-label classification datasets, MedFM-Chest and MedFM-Endo, although the expected overlap is not observed, the distance is closer compared to FULL. These results indicate that pretrained representations can be preserved without damaging their semantic structure by imposing orthogonal constraints, upon which new branches for downstream tasks can grow. This may unlock a temporal view of the Platonic Representation Hypothesis, This may unlock a temporal view of the Platonic Representation Hypothesis, suggesting that models have sufficient capacity to accommodate pretrained and finetuned knowledge simultaneously, yielding a shared representation.

Table 2: **Quantitative result on different backbones**. We adopt large-scale ViT pretrained on ImageNet-21K with supervision and DINOv2 ViT-base.

| Model | Single-label datasets | | | | | | Multi-label datasets | | | | Average | |
|---|---|---|---|---|---|---|---|---|---|---|---|---|
| | ISIC2018 (7) | | APTOS2019 (5) | | MedFM-Colon (2) | | MedFM-Chest (19) | | MedFM-Endo (4) | | | |
| | Acc | Sim | Acc | Sim | Acc | Sim | mAP | Sim | mAP | Sim | Acc/mAP | Sim |
| **ViT-large(ImageNet-21K)** | | | | | | | | | | | | |
| Scratch | 66.07 | 0.46 | 69.67 | 0.40 | 89.88 | 0.59 | 12.12 | 0.47 | 17.30 | 0.34 | 51.01 | 0.45 |
| Linear | 78.44 | 1.00 | 78.69 | 1.00 | 95.19 | 1.00 | 25.13 | 1.00 | 38.11 | 1.00 | 63.11 | 1.00 |
| FULL | 83.33 | 0.56 | 83.61 | 0.88 | **99.71** | 0.69 | 35.39 | 0.53 | 55.58 | **0.67** | 71.52 | 0.67 |
| +RIFT (ours) | 83.66 | **0.60** | 82.24 | **0.89** | 99.12 | **0.73** | 34.39 | **0.81** | 58.23 | 0.66 | +0.01% | +11.05% |
| +RIFT* (ours) | **84.13** | 0.52 | **83.06** | 0.87 | 99.51 | 0.75 | **38.06** | 0.67 | **58.23** | 0.66 | +1.50% | +4.56% |
| **ViT-base(DINOv2 Oquab et al. (2023))** | | | | | | | | | | | | |
| Scratch | 60.58 | 0.22 | 54.37 | 0.65 | 85.36 | 0.69 | 10.46 | 0.78 | 16.08 | 0.37 | 45.37 | 0.54 |
| Linear | 75.53 | 1.00 | 80.33 | 1.00 | 94.06 | 1.00 | 23.24 | 1.00 | 36.12 | 1.00 | 61.85 | 1.00 |
| FULL | 76.82 | 0.30 | **76.78** | 0.49 | 98.13 | 0.53 | 12.29 | **0.64** | 22.20 | **0.37** | 57.24 | 0.47 |
| +RIFT (ours) | 69.78 | 0.40 | 76.50 | 0.52 | 92.34 | **0.68** | 12.83 | 0.62 | 38.18 | 0.35 | +1.19% | +10.30% |
| +RIFT* (ours) | **79.63** | 0.45 | 76.50 | **0.52** | **99.71** | 0.48 | 12.83 | 0.62 | **40.15** | 0.36 | +7.89% | +4.39% |

**Quantitative result on different backbones**. Tab. 2 presents the results of RIFT applied to ViT-large and DINOv2 backbones. When applied to larger models and the self-supervised DINOv2 backbone, RIFT consistently improves representation similarity while maintaining, or in some cases slightly improving, classification performance. Notably, larger models overall exhibit higher post-finetuning representation similarity, indicating stronger representational stability. In addition, the self-supervised pretrained DINOv2 model shows that finetuning only the linear head (LINEAR) outperforms full finetuning (FULL), especially on the MedFM-Chest dataset where the gap is substantial, highlighting the superior generalization ability of self-supervised pretraining.

### 5.3 GENERALIZATION RESULT

Table 3: **Qualitative result on zero-shot natural image classification.** We adopt model finetuned on ISIC2018 with several classical natural image datasets Parkhi et al. (2012); Nilsback & Zisserman (2008); Netzer et al. (2011); Krause et al. (2013); Wah et al. (2011) estimated using 20-NN. RIFT* indicates a regularization coefficient of $\lambda = 0.6$.

| Model | Oxford-IIIT Pet | Oxford Flowers | SVHN | Stanford Cars | CUB-200 | Avg |
|---|---|---|---|---|---|---|
| Pretrained Backbone | 84.19 | **96.93** | 38.39 | 25.50 | 54.96 | 59.99 |
| FULL (Sim=0.32) | 82.03 | 90.41 | 38.36 | 26.03 | 55.10 | 58.39 |
| +RIFT (Sim=0.60) | 83.65 | 95.44 | 37.83 | 25.61 | 55.29 | 59.56 |
| +RIFT* (Sim=0.48) | **84.38** | 95.38 | **39.59** | **27.11** | 55.29 | **60.35** |

**Qualitative result on zero-shot natural image classification.** As shown in Table 3, we performed KNN evaluation on five unseen datasets to compare the generalization ability of different finetuned models. Supporting our hypothesis, the pretrained models generally exhibit better overall generalization performance than the model fully finetuned on the ISIC2018 dataset (FULL). Furthermore, compared to FULL, models with higher similarity (RIFT and RIFT*) tend to better preserve the generalization capability of the pretrained models. Interestingly, RIFT and RIFT* even improves generalization, likely due to incorporating additional transferable patterns in finetuning data.

**Quantitative result on theoretical generalizability.** Tab. 4 evaluates the generalizability of finetuned models by comparing the sharpness of the loss function (i.e., stability under perturbations),

Table 4: **Quantitative result on theoretical generalizability.** We validate sharpness $\max_{\|\epsilon\| \leq \rho} \mathcal{L}_{\text{cls}}(\theta + \epsilon) - \mathcal{L}_{\text{cls}}(\theta)$ with lower values indicating better generalization across methods: FULL, RIFT($\lambda = 1$), and LINEAR, with $\rho = 0.01$.

| Metric | Model | ISIC2018 | APTOS2019 | MedFM-Chest | MedFM-Colon | MedFM-Endo | Avg. |
|---|---|---|---|---|---|---|---|
| $\max_{\|\epsilon\| \leq \rho} \mathcal{L}_{\text{cls}}(\theta + \epsilon) \downarrow$ | LINEAR | 0.7292 | 0.6442 | 4.7553 | 0.1995 | 1.1375 | 1.4931 |
| | FULL | 0.6398 | 0.5701 | 4.3836 | **0.0509** | 1.0859 | 1.3461 |
| | RIFT (ours) | **0.5558** | **0.5262** | **4.3338** | 0.0677 | **1.0036** | **1.2974** |
| $\mathcal{L}_{\text{cls}}(\theta) \downarrow$ | LINEAR | 0.7159 | 0.5989 | 4.7415 | 0.1841 | 1.1343 | 1.4749 |
| | FULL | 0.6125 | **0.5049** | 4.3727 | **0.0305** | 1.0743 | 1.3190 |
| | RIFT (ours) | **0.5312** | 0.5082 | **4.3236** | 0.0496 | **0.9957** | **1.2817** |
| Sharpness $\downarrow$ | LINEAR | **0.0133** | 0.0453 | **0.0138** | 0.0154 | **0.0032** | 0.0182 |
| | FULL | 0.0273 | 0.0652 | 0.0109 | 0.0204 | 0.0116 | 0.0271 |
| | RIFT (ours) | 0.0246 | **0.0180** | **0.0102** | 0.0181 | **0.0080** | **0.0158** |

measured as $\max_{\|\epsilon\| \leq \rho} \mathcal{L}_{\text{cls}}(\theta + \epsilon) - \mathcal{L}_{\text{cls}}(\theta)$, across different finetuning methods. Lower sharpness reflects better generalizability. LINEAR outperforms FULL with an average sharpness of 0.0182, demonstrating its robustness due to the pretrained initialization. FULL exhibits a higher average sharpness of 0.0271, indicating weaker generalization. RIFT achieves the lowest average sharpness of 0.0158, demonstrating its ability to achieve flatter minima and better generalization. Notably, RIFT consistently performs better than FULL across all datasets, with particularly strong results on APTOS2019, where its sharpness (0.0180) is significantly lower than that of FULL (0.0652). These results highlight that, under representation similarity constraints, RIFT actively guides the optimization landscape toward flatter and more generalizable minima.

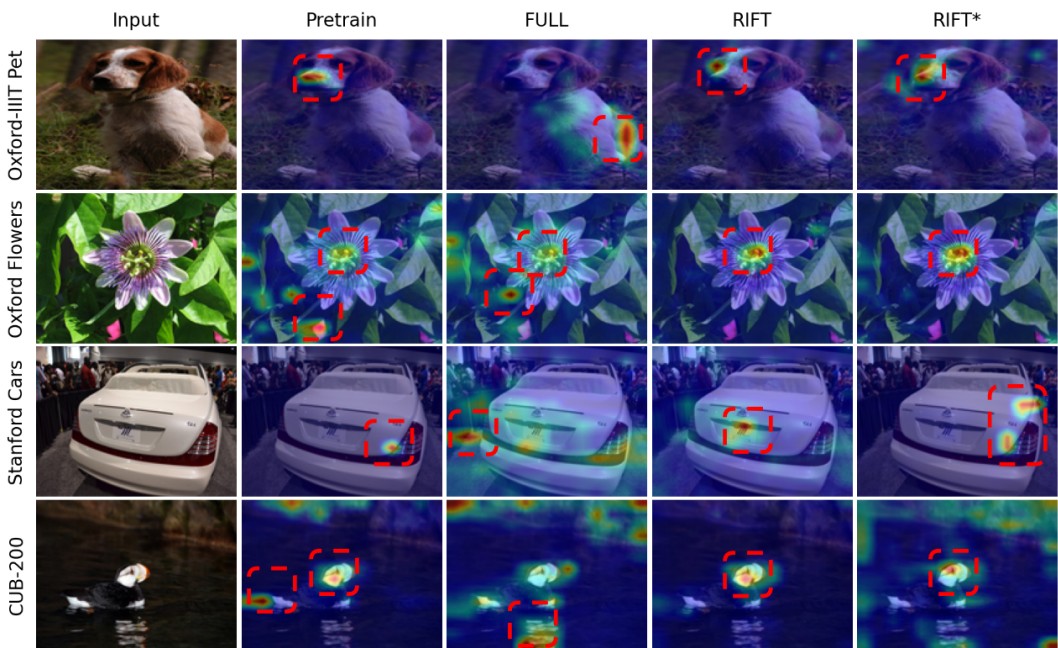

Figure 5: **Attention heatmap visualization.** We give more qualitative results on zero-shot natural image classification to further demonstrate the generalizability of RIFT and RIFT*. The red boxes highlight the regions most attended to by each method. Images are taken from previously unseen datasets: Oxford-IIIT Pet, Oxford Flowers, Stanford Cars, and CUB-200.

**Attention heatmap visualization.** As shown in Fig. 5, the pretrained model focuses on the nose and mouth regions of dogs in Oxford-IIIT Pet and on the head regions of birds in CUB-200. In contrast, after direct finetuning (FULL), the attention shifts to less relevant regions, such as the back of the dog or the bird's reflection in the water. This supports the observation that pretrained and finetuned (FULL) models generalize differently on unseen datasets, although both tend to overlook the flower centers in the Oxford Flowers dataset. RIFT and RIFT, however, retain the generalization patterns of

pretrained representations, correctly attending to class-relevant objects across all cases, and in some instances even improving upon them (e.g., on the flower dataset), thereby further strengthening the generalization ability of current learned representations.

## 5.4 ABLATION STUDY

Ablation studies are conducted to further explore the impact of different components. The key differences among the examined methods lie in whether the $Q$ matrix is learnable, the application of mean alignment (*i.e.*, using $\mu$), the chosen batch size and the regularization coefficient $\lambda$.

**Ablation of matrix Q**. As shown in Tab. 5 (a), employing a learnable $Q$ matrix improves accuracy and similarity metrics by $0.4\%$ and $0.04$, respectively, compared to using a fixed $Q$ parameter. This demonstrates that a learnable orthogonal matrix can more effectively preserve pretrained representations and leverage their benefits to strengthen downstream task adaptation.

Table 5: **Ablation study on fixed Q, aligning $\mu$, and batch size.**

(a) Ablation of matrix Q.

| Matrix Q | Acc | Sim |
|---|---|---|
| Fixed Q | 83.55 | 0.56 |
| Learnable Q | 83.95↑ | 0.60↑ |

(b) Ablation of aligning $\mu$.

| mean $\mu$ | Acc | Sim |
|---|---|---|
| w/ $\mu$ | 83.49 | 0.51 |
| w/o $\mu$ | 83.95↑ | 0.60↑ |

(c) Ablation of batch size.

| Batch Size | Acc | Sim |
|---|---|---|
| 4 | 81.42 | 0.21 |
| 32 | 83.33 | 0.30 |
| 128 | 83.95↑ | 0.60↑ |

**Ablation of alignment of $\mu$**. Tab. 5 (b) shows that removing mean alignment (w/o $\mu$) achieves better accuracy and similarity, with gains of $0.46\%$ and $0.09$ over applying it (w/ $\mu$). This indicates that mean alignment may impose unnecessary constraints on feature distribution, limiting the model's ability to exploit data structure. Without it, the model learns more flexible representations, yielding improved performance.

**Ablation of batch size**. Increasing the batch size from 4 to 128, as shown in Tab. 5 (c), significantly boosts accuracy and similarity metrics, with improvements of $2.53\%$ and $0.39$, respectively. A larger batch size pro-

Table 6: **Ablation study of $\lambda$ on ISIC2018 dataset.**

| $\lambda$ | 0.1 | 0.2 | 0.4 | 0.6 | 0.8 | 1.0 |
|---|---|---|---|---|---|---|
| Acc ↑ | 84.82 | 84.57 | 84.33 | 84.85 | 83.73 | 83.95 |
| Sim ↑ | 0.35 | 0.38 | 0.49 | 0.48 | 0.53 | 0.60 |

vides a more accurate estimation of the covariance matrix. The observed trend underscores the importance of selecting an appropriate batch size.

**Ablation of $\lambda$**. As shown in Table 6, we examine the effect of the regularization coefficient $\lambda$ on different datasets and finetuning methods, varying it from 0.1 to 1.0. Overall, choosing an appropriate $\lambda$ yields dual benefits of adaptation and generalization. Even under extreme settings (*e.g.*, $\lambda = 1$), downstream performance shows only a slight drop without sacrificing the effective adaptation. Additional results are provided in Tab. A2 and Tab. A3 in the Appendix.

## 6 CONCLUSION

We propose *Representation Invariance FineTuning (RIFT)*, a simple and efficient constraint that preserves pretrained representations by enforcing orthogonal-invariant CKA similarity between pretrained and finetuned models. Our experiments across cross-domain and low-resource scenarios show that RIFT integrates seamlessly with mainstream finetuning methods, achieving competitive or even improved downstream task performance while mitigating the loss of generalization. These findings suggest that effective adaptation does not need to come at the cost of established semantic knowledge and generalizability, and highlight the value of explicitly preserving representation invariance during finetuning. We believe this work opens up promising directions for finetuning paradigms that emphasize the compatibility of adaptation and generalization in foundation models.

**Limitations.** Although our work demonstrates the effectiveness of RIFT in vision foundation models, the challenge of preserving pretrained representations extends beyond vision. Future research should investigate its applicability to other modalities (e.g., large language and multimodal models) and broader cross-domain or low-resource tasks, such as embodied AI, mathematics, and programming.

ETHICS STATEMENT

This work complies with the Code of Ethics. Our research does not involve human subjects, sensitive personal data, or experiments that could raise ethical concerns. No hical issues are associated with the methods, experiments, or results presented in this paper.

REPRODUCIBILITY STATEMENT

Hyperparameter configurations are detailed in Section 5 and Appendix A1. We also intend to release the code and model checkpoints to enable full reproduction of the results reported in this paper.

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

APPENDIX

LLM USAGE STATEMENT

ChatGPT-5 are used only as general-purpose assistive tools, such as for language polishing and improving readability. No part of the research ideation, experimental design, or substantive writing is generated by LLMs. The authors take full responsibility for the content of this paper.

## A1 EXPERIMENT SETTINGS

We use the ViT-B/16 model Dosovitskiy et al. (2020), with an input image size of $224 \times 224$ and a patch size of 16. The pretrained model is trained in a supervised manner on ImageNet-1K. In our experiments, the batch size is set to 128 to obtain a better estimation of the covariance matrix of the finetuned features, and the learning rate is set to $6 \times 10^{-4}$. The experiments were conducted on 4 NVIDIA A100 40G GPUs, with each training session running for 50 epochs. Our code is built upon the MMClassification framework Mmc. Tab. A1 presents detailed information about the experimental dataset.

For Tab. 2, we use the ViT-Large model pretrained on ImageNet-21K in a supervised manner and the DINOv2 (Base) model pretrained in a self-supervised manner. Their patch sizes are 16 and 14, input image sizes are $384 \times 384$ and $518 \times 518$, and feature dimensions are 1024 and 768, respectively.

Table A1: Details of the dataset specifications.

| Dataset | Modality | Task Type | Classes | Train | Test | Metric |
|---|---|---|---|---|---|---|
| ISIC2018 Codella et al. (2019) | Dermoscopy | Multiclass | 7 | 10,015 | 1,512 | Accuracy |
| APTOS2019 Karthik & Dane | Fundus | Multiclass | 5 | 2,930 | 366 | Accuracy |
| MedFM-Chest Wang et al. (2023) | X-ray | Multi-label | 19 | 2,140 | 3,869 | mAP |
| MedFM-Colon Wang et al. (2023) | Pathology | Binary | 2 | 5,654 | 7,651 | Accuracy |
| MedFM-Endo Wang et al. (2023) | Endoscopy | Multi-label | 4 | 1,810 | 2,936 | mAP |

## A2 MORE RESULTS

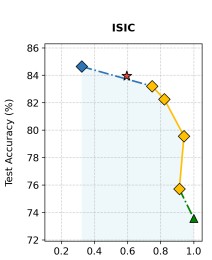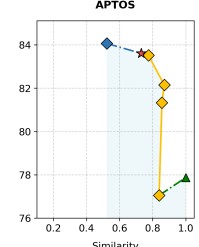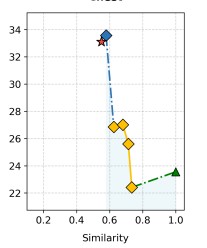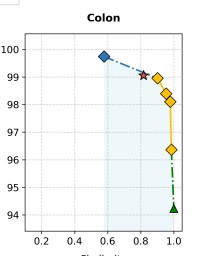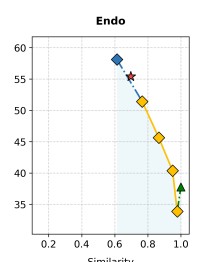

Figure A1: Similarity–accuracy curves of the representation similarity constrained (RSC) method across different datasets. Models above the curve achieve a comparable balance between similarity preservation and accuracy. The $\lambda$ values range from $4.0$ to $0.3$, indicating a gradual decrease in the weight of the similarity loss.

**Similarity and accuracy curve.** As shown in Fig. A1, our proposed RIFT method consistently demonstrates an effective balance between similarity (sim) and accuracy (acc) across multiple datasets. Direct finetuning, the RSC method ($\lambda$ ranges from 4.0 to 0.3), and finetuning only the linear classification head form the trade-off curve between classification accuracy and representation similarity (with pretrained model). Models above the curve indicate a better balance between accuracy and similarity, while those below the curve show the opposite. Direct finetuning achieves the highest classification accuracy but the lowest representation similarity. Additionally, the trade-off curve

Table A2: Detailed evaluation results of RIFT and RSC with different regularization coefficients $\lambda$ of Tab. 1. The selected $\lambda$ corresponds to the parameter yielding the highest accuracy/mAP.

| coefficient $\lambda$ | Single-label datasets | | | | | | Multi-label datasets | | | |
|---|---|---|---|---|---|---|---|---|---|---|
| | ISIC2018 (7) | | APTOS2019 (5) | | MedFM-Colon (2) | | MedFM-Chest (19) | | MedFM-Endo (4) | |
| | Acc | Sim | Acc | Sim | Acc | Sim | mAP | Sim | mAP | Sim |
| **RIFT Method** | | | | | | | | | | |
| $\lambda = 1.0$ | 83.95 | 0.60 | 83.61 | 0.73 | 99.05 | 0.82 | 33.11 | 0.55 | 55.42 | 0.70 |
| $\lambda = 0.8$ | 83.73 | 0.53 | 84.43 | 0.76 | 99.12 | 0.84 | 32.22 | 0.59 | 55.16 | 0.70 |
| $\lambda = 0.6$ | **84.85** | 0.48 | **85.25** | 0.54 | **99.51** | 0.84 | **35.24** | 0.62 | 53.66 | 0.72 |
| $\lambda = 0.4$ | 84.33 | 0.50 | 84.70 | 0.71 | 99.51 | 0.83 | 34.01 | 0.53 | 56.45 | 0.67 |
| $\lambda = 0.2$ | 84.57 | 0.38 | 83.47 | 0.53 | 99.44 | 0.82 | 34.47 | 0.56 | 57.38 | 0.67 |
| $\lambda = 0.1$ | 84.82 | 0.35 | 84.84 | 0.59 | 99.41 | 0.83 | 34.02 | 0.44 | **58.62** | 0.66 |
| **RSC Method** | | | | | | | | | | |
| $\lambda = 4.0$ | 75.71 | 0.91 | 77.05 | 0.84 | 96.37 | 0.98 | 22.41 | 0.73 | 33.88 | 0.98 |
| $\lambda = 1.5$ | 79.56 | 0.94 | 81.33 | 0.86 | 98.10 | 0.98 | 25.60 | 0.71 | 40.38 | 0.95 |
| $\lambda = 0.7$ | 82.25 | 0.82 | 82.15 | 0.87 | 98.40 | 0.95 | 27.01 | 0.68 | 45.63 | 0.87 |
| $\lambda = 0.3$ | 83.20 | 0.75 | 83.52 | 0.77 | 98.95 | 0.90 | 26.86 | 0.62 | 51.41 | 0.77 |
| $\lambda = 0.2$ | 83.47 | 0.35 | 82.79 | 0.53 | **99.51** | 0.82 | 33.73 | 0.56 | **57.52** | 0.67 |
| $\lambda = 0.1$ | **85.19** | 0.35 | **84.97** | 0.59 | 99.12 | 0.83 | **34.62** | 0.44 | 56.26 | 0.66 |
| **LoRA+RIFT Method** | | | | | | | | | | |
| $\lambda = 1.0$ | 77.51 | 0.74 | 79.24 | 0.79 | 24.28 | 0.63 | 96.07 | 0.82 | 39.24 | 0.76 |
| $\lambda = 0.2$ | 78.84 | 0.60 | 80.05 | 0.73 | 25.51 | 0.59 | 96.07 | 0.81 | 39.69 | 0.77 |
| $\lambda = 0.1$ | 80.32 | 0.46 | 80.19 | 0.67 | **25.77** | 0.61 | **96.37** | 0.79 | 38.68 | 0.78 |
| $\lambda = 0.02$ | **80.92** | 0.40 | **81.01** | 0.61 | 24.42 | 0.73 | 96.17 | 0.80 | **39.88** | 0.77 |
| **Adaptformer+RIFT Method** | | | | | | | | | | |
| $\lambda = 1$ | 77.65 | 0.74 | 80.15 | 0.85 | 23.82 | 0.59 | 95.87 | 0.86 | 38.63 | 0.75 |
| $\lambda = 0.1$ | 79.17 | 0.64 | **80.60** | 0.81 | 24.97 | 0.64 | 96.07 | 0.80 | **40.55** | 0.71 |
| $\lambda = 0.02$ | **79.96** | 0.56 | 79.51 | 0.79 | **25.24** | 0.65 | **97.54** | 0.81 | 37.16 | 0.61 |
| **VPT+RIFT Method** | | | | | | | | | | |
| $\lambda = 1$ | 76.32 | 0.74 | 78.69 | 0.85 | 22.19 | 0.59 | **95.48** | 0.86 | 37.70 | 0.75 |
| $\lambda = 0.1$ | 77.38 | 0.68 | **78.96** | 0.66 | 24.27 | 0.56 | 94.50 | 0.74 | 37.68 | 0.71 |
| $\lambda = 0.02$ | **78.31** | 0.50 | 78.69 | 0.57 | **24.58** | 0.47 | 95.38 | 0.72 | **40.54** | 0.67 |

Table A3: Detailed evaluation results of RIFT with different regularization coefficients $\lambda$ from Tab. 2. The selected $\lambda$ corresponds to the parameter yielding the highest accuracy/mAP.

| coefficient $\lambda$ | Single-label datasets | | | | | | Multi-label datasets | | | |
|---|---|---|---|---|---|---|---|---|---|---|
| | ISIC2018 (7) | | APTOS2019 (5) | | MedFM-Colon (2) | | MedFM-Chest (19) | | MedFM-Endo (4) | |
| | Acc | Sim | Acc | Sim | Acc | Sim | mAP | Sim | mAP | Sim |
| **ViT-large(ImageNet-21K)+RIFT Method** | | | | | | | | | | |
| $\lambda = 1.0$ | 83.66 | 0.60 | 82.24 | 0.89 | 34.39 | 0.81 | 99.12 | 0.73 | **58.23** | 0.66 |
| $\lambda = 0.1$ | **84.13** | 0.52 | **83.06** | 0.87 | **38.06** | 0.67 | **99.51** | 0.75 | 56.40 | 0.68 |
| **ViT-base(Dinov2)+RIFT Method** | | | | | | | | | | |
| $\lambda = 1.0$ | 69.78 | 0.40 | **76.50** | 0.52 | **12.83** | 0.62 | 92.34 | 0.68 | 38.18 | 0.35 |
| $\lambda = 0.1$ | **79.63** | 0.45 | 68.31 | 0.28 | 12.76 | 0.67 | **99.71** | 0.48 | **40.15** | 0.36 |

highlights the rapid decline in accuracy as similarity increases (with higher weight on RSC). Our proposed RIFT method shows comparable results on the ISIC2018, APTOS2019, MedFM-Colon, and MedFM-Endo datasets compared to the RSC. Although on the MedFM-Chest dataset, we failed to find a balance in the similarity-accuracy trade-off, these overall results strongly demonstrate that RIFT

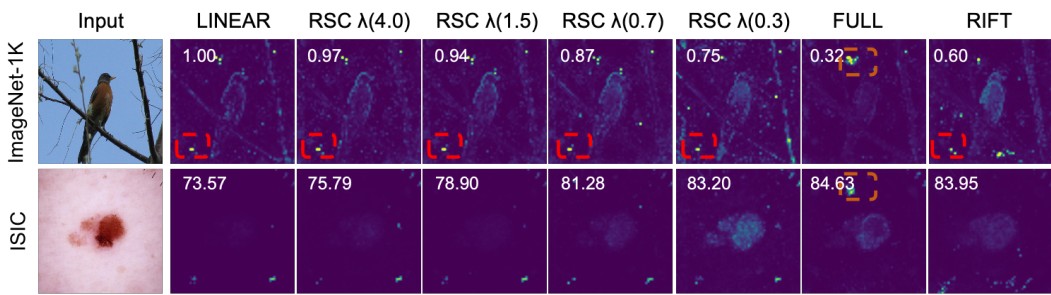

Figure A2: Visualization of attention maps for different finetuning methods on the ISIC2018 dataset. The first row illustrates how the finetuned models forget original knowledge as representation similarity changes, while the second row shows how they learn the new task at the corresponding similarity levels. **This also highlights the potential of RIFT for future applications in continual and multi-task learning.**

Table A4: Comparison of parameter distance to average center for FULL and RIFT methods across five datasets.

| Dataset | FULL | RIFT | Change↓ |
|---|---|---|---|
| ISIC2018 | 1.35 | 0.93 | ↓ 0.42 |
| APTOS2019 | 0.49 | 0.45 | ↓ 0.04 |
| MedFM-Chest | 1.02 | 0.98 | ↓ 0.04 |
| MedFM-Colon | 0.45 | 0.67 | ↑ 0.22 |
| MedFM-Endo | 0.68 | 0.58 | ↓ 0.10 |
| Avg. | 0.80 | 0.72 | ↓ 0.08 |

effectively leverages the trade-off between similarity and accuracy, avoiding the severe similarity degradation observed in the direct finetuning method (FULL) when enforcing high accuracy, thereby achieving superior overall performance.

**Visualization of attention maps demonstrating learning and forgetting.** As shown in Fig. A2, the first row illustrates the changes in the attention maps of the pretrained model as similarity decreases. After finetuning directly on the ISIC2018 dataset (FULL), the attention to the ImageNet-1K image target (bird) vanishes. In contrast, applying a direct similarity constraint (RSC) or an indirect constraint (RIFT) preserves attention to the target object, even retaining artifacts (red dashed box). The second row shows that as classification accuracy improves, the finetuned model gradually enhances its attention to the target object (skin lesion). Notably, the artifacts from the pretrained model are also preserved. For FULL finetuning, the artifacts (orange dashed box) appear in both ImageNet-1K and ISIC2018 images. These observations highlight the necessity of considering similarity constraints.

**Enhanced parameter centrality for multi-task learning.** Results in Tab. A4 demonstrate that the RIFT method exhibits better parameter centrality across multiple datasets, thereby validating its adaptability in multi-task training. Specifically, the RIFT method significantly reduces the distance of model parameters to the average parameter center on four datasets: ISIC2018, APTOS2019, MedFM-Chest, and MedFM-Endo, with reductions of 0.42, 0.04, 0.04, and 0.10, respectively. Although the distance increases slightly (by 0.22) on the MedFM-Colon dataset, the overall performance of RIFT remains superior to the FULL method, with an average distance reduction of 0.08. These results indicate that the RIFT method can more effectively extract shared features, achieving better parameter centrality and generalization in multi-task training.

**Training time comparison.** The results in Tab. A5 reveal significant variations in training efficiency across the evaluated methods. The LINEAR method demonstrates the fastest performance, requiring the least time for both per-iteration and full-epoch training. In contrast, the FULL and RIFT methods exhibit nearly identical computational costs, suggesting comparable efficiency. Notably, the RSC

Table A5: Training time per iteration and for 50 epochs on single NVIDIA A100 40G with a batch size of 128.

| Metric | LINEAR | FULL | RSC | RIFT |
|---|---|---|---|---|
| One Iteration (s) | 3.91 | 8.38 | 16.68 | 8.47 |
| 50 Epochs (min) | 26.65 | 57.08 | 111.04 | 57.83 |

method incurs the highest computational overhead, making it the least efficient among the tested approaches.

**Computational Complexity Analysis.** The loss in Eq. equation 11 promotes the transferability of mean vectors while preserving the structural invariance of covariance matrices under orthogonal transformations. The orthonormal constraint $\mathbf{Q} \in \mathcal{V}_{d,d}$ guarantees that rotations in the feature space remain isometric. By decoupling mean and covariance alignment, the computational complexity is reduced from $O(nd^2)$ to $O(d^2)$ through trace-based operations, while still maintaining the geometric structure of the representation space. This simplification preserves the essential properties enforced by orthogonal covariance transformations without introducing additional approximation error.

## A3 ANALYSIS

**Definition A3.1** (Multi-layer linear network with layerwise orthogonal rotations). *Let $X \in \mathbb{R}^{n \times d}$ be the input data with $n$ samples and $d$ features, and let $W_1, W_2, \ldots, W_L \in \mathbb{R}^{d \times d}$ be the weight matrices of an $L$-layer linear network. Define the network*

$$f(X) = XW_1W_2 \cdots W_L \in \mathbb{R}^{n \times d} \tag{A1}$$

*Consider applying layerwise orthogonal rotations $Q_1, Q_2, \ldots, Q_L \in O(d)$ to obtain the rotated network*

$$f'(X) = X(W_1Q_1)(W_2Q_2) \cdots (W_LQ_L) = XW' \tag{A2}$$

*where $O(d)$ denotes the set of $d \times d$ orthogonal matrices. Denote by $\sigma_1(\cdot)$ the largest singular value (spectral norm) of a matrix.*

**Assumption A3.2.** *Let $W = W_1W_2 \cdots W_L \in \mathbb{R}^{d \times d}$ denote the weight matrix of a pretrained $L$-layer linear network. We assume that $W$ exhibits strong generalization, formalized by*

$$\sigma_1(W) \ll \min_{i=1,\ldots,L} \sigma_1(W_i) \tag{A3}$$

**Proposition A3.3** (Informal). *Let $W = W_1W_2 \cdots W_L$ be an $L$-layer linear network with $W_i \in \mathbb{R}^{d \times d}$, and assume that $W$ exhibits strong generalization in the sense of Assumption A3.2, i.e.,*

$$\sigma_1(W) \ll \min_{i=1,\ldots,L} \sigma_1(W_i) \tag{A4}$$

*Then there exist orthogonal matrices $Q_1, \ldots, Q_L \in O(d)$ such that the rotated network*

$$W' = W_1Q_1W_2Q_2 \cdots W_LQ_L \tag{A5}$$

*has a strictly larger spectral norm than the original $W$.*

*Intuitively, because the pretrained network already generalizes well, its layers are not perfectly aligned to maximize the spectral norm, so an appropriate choice of layerwise rotations can increase it by improving the alignment of dominant singular directions across layers.*

**Assumption A3.4** (Bounded Cross-Covariance). *Let $X$ with $n$ samples and feature dimension $d$, and denote the feature matrices $Y = F_\theta(X) \in \mathbb{R}^{n \times d}$, $Y_0 = F_{\theta_0}(X) \in \mathbb{R}^{n \times d}$. Let $Q \in \mathbb{R}^{d \times d}$ be orthogonal ($Q^\top Q = I_d$) and $\alpha \in \mathbb{R}$, and define the aligned feature matrix $Z := \alpha Y_0 Q$ with residual $\varepsilon := Y - Z$.*

*For any $A \in \mathbb{R}^{n \times d}$, let $\bar{A} := \frac{1}{n} 1_n^\top A \in \mathbb{R}^{1 \times d}$ denote its column mean, where $1_n \in \mathbb{R}^n$ is the all-ones vector, and define the column-centered version $\widetilde{A} := A - 1_n \bar{A}$. The empirical cross-covariance between $A, B \in \mathbb{R}^{n \times d}$ is given by $\mathrm{Cov}(A, B) := \frac{1}{n} \widetilde{A}^\top \widetilde{B} \in \mathbb{R}^{d \times d}$.*

*By construction, $\mathrm{Cov}(\varepsilon, Y_0) = \mathrm{Cov}(Y, Y_0) - \alpha Q^\top \mathrm{Cov}(Y_0, Y_0)$. We assume this cross-covariance is bounded as $\|\mathrm{Cov}(\varepsilon, Y_0)\|_F \leq \gamma$ for some constant $\gamma \geq 0$.*

**Theorem A3.5.** *Define* $Y := F_\theta(X) \in \mathbb{R}^{n \times d}$, $Y_0 := F_{\theta_0}(X) \in \mathbb{R}^{n \times d}$ *as feature matrices of* $n$ *samples with* $d$-*dimensional features. Denote the row-wise means* $\mu_\theta := \frac{1}{n}1_n^\top Y$, $\mu_{\theta_0} := \frac{1}{n}1_n^\top Y_0$, *and the empirical covariances* $\Sigma_\theta := \frac{1}{n}(Y - 1_n\mu_\theta)^\top(Y - 1_n\mu_\theta)$, $\Sigma_{\theta_0} := \frac{1}{n}(Y_0 - 1_n\mu_{\theta_0})^\top(Y_0 - 1_n\mu_{\theta_0})$. *Let* $Q \in \mathbb{R}^{d \times d}$ *be orthogonal* ($Q^\top Q = I_d$) *and* $\alpha \in \mathbb{R}$. *Define* $Z := \alpha Y_0 Q$, $\varepsilon := Y - Z$. *Assume the cross-covariance* $\mathrm{Cov}(\varepsilon, Y_0) := \frac{1}{n}(\varepsilon - 1_n\bar\varepsilon)^\top(Y_0 - 1_n\bar Y_0)$ *satisfies* $\|\mathrm{Cov}(\varepsilon, Y_0)\|_F \leq \gamma$, *where* $\bar\varepsilon := \frac{1}{n}1_n^\top\varepsilon$, $\bar Y_0 := \frac{1}{n}1_n^\top Y_0$. *Define* $\mathcal{E} := \frac{1}{n}\|Y - \alpha Y_0 Q\|_F^2$, $\Delta\mu := \mu_\theta - \alpha\mu_{\theta_0}Q$, $\Delta\Sigma := \Sigma_\theta - \alpha^2 Q^\top\Sigma_{\theta_0}Q$. *Then*

$$\left|\mathcal{E} - \left(\|\Delta\mu\|_2^2 + \mathrm{tr}(\Delta\Sigma)\right)\right| \leq 2|\alpha|\sqrt{d}\,\gamma \tag{A6}$$

*Proof.* Let

$$\varepsilon = Y - Z = Y - \alpha Y_0 Q \tag{A7}$$

Its row-wise mean is

$$\bar\varepsilon = \frac{1}{n}1_n^\top\varepsilon = \mu_\theta - \alpha\mu_{\theta_0}Q = \Delta\mu \tag{A8}$$

Define the centered matrices

$$\widetilde Y = Y - 1_n\mu_\theta, \quad \widetilde Y_0 = Y_0 - 1_n\mu_{\theta_0}, \quad \widetilde\varepsilon = \varepsilon - 1_n\bar\varepsilon = \widetilde Y - \alpha\widetilde Y_0 Q \tag{A9}$$

so that

$$\mathrm{Cov}(\varepsilon) = \frac{1}{n}\widetilde\varepsilon^\top\widetilde\varepsilon \tag{A10}$$

Using $Y = Z + \varepsilon$ and bilinearity of covariance, we have

$$\mathrm{Cov}(Y) = \mathrm{Cov}(Z) + \mathrm{Cov}(\varepsilon) + \mathrm{Cov}(\varepsilon, Z) + \mathrm{Cov}(Z, \varepsilon) \tag{A11}$$

Since $\widetilde Z = \alpha\widetilde Y_0 Q$, it follows that

$$\mathrm{Cov}(Z) = \frac{1}{n}\widetilde Z^\top\widetilde Z = \alpha^2 Q^\top\Sigma_{\theta_0}Q \tag{A12}$$

Therefore

$$\Delta\Sigma = \mathrm{Cov}(Y) - \mathrm{Cov}(Z) = \mathrm{Cov}(\varepsilon) + \mathrm{Cov}(\varepsilon, Z) + \mathrm{Cov}(Z, \varepsilon) \tag{A13}$$

and hence

$$\mathrm{tr}(\Delta\Sigma) = \mathrm{tr}(\mathrm{Cov}(\varepsilon)) + 2\,\mathrm{tr}(\mathrm{Cov}(\varepsilon, Z)) \tag{A14}$$

On the other hand, the error can be written as

$$\mathcal{E} = \frac{1}{n}\|Y - \alpha Y_0 Q\|_F^2 = \frac{1}{n}\|\varepsilon\|_F^2 = \|\Delta\mu\|_2^2 + \mathrm{tr}(\mathrm{Cov}(\varepsilon)) \tag{A15}$$

Subtracting $\|\Delta\mu\|_2^2 + \mathrm{tr}(\Delta\Sigma)$ gives

$$\mathcal{E} - \left(\|\Delta\mu\|_2^2 + \mathrm{tr}(\Delta\Sigma)\right) = -2\,\mathrm{tr}(\mathrm{Cov}(\varepsilon, Z)) \tag{A16}$$

Now observe that

$$\mathrm{Cov}(\varepsilon, Z) = \frac{1}{n}\widetilde\varepsilon^\top\widetilde Z = \frac{1}{n}\widetilde\varepsilon^\top(\alpha\widetilde Y_0 Q) = \alpha\,\mathrm{Cov}(\varepsilon, Y_0)Q \tag{A17}$$

Therefore

$$|\mathrm{tr}(\mathrm{Cov}(\varepsilon, Z))| = |\alpha|\,|\mathrm{tr}(\mathrm{Cov}(\varepsilon, Y_0)Q)| \leq |\alpha|\sqrt{d}\,\|\mathrm{Cov}(\varepsilon, Y_0)\|_F \leq |\alpha|\sqrt{d}\,\gamma \tag{A18}$$

where we used $|\mathrm{tr}(AQ)| \leq \|A\|_* \leq \sqrt{d}\,\|A\|_F$

Combining the results, we conclude

$$\left|\mathcal{E} - (\|\Delta\mu\|_2^2 + \mathrm{tr}(\Delta\Sigma))\right| = 2\,|\mathrm{tr}(\mathrm{Cov}(\varepsilon, Z))| \leq 2|\alpha|\sqrt{d}\,\gamma \tag{A19}$$

$\square$

**Remark A3.6** (Interpretation and Significance). *Theorem A3.5 shows that the RIFT objective, expressed in terms of the mean difference* $\Delta\mu$ *and the covariance difference* $\Delta\Sigma$, *accurately approximates the original alignment loss* $\mathcal{E}$. *Specifically, if the residual* $\varepsilon$ *has zero cross-covariance with the pretrained representation* $Y_0$ (*i.e.,* $\gamma = 0$), *the RIFT objective equals the true alignment loss exactly. When small cross-covariances exist, the discrepancy is explicitly bounded by* $2|\alpha|\sqrt{d}\,\gamma$, *depending only on the feature dimension and the cross-covariance magnitude. This provides theoretical support for using covariance-based orthogonal alignment: it eliminates the need to compute* $\mathcal{E}$ *over all samples while retaining the key alignment characteristics of the original representation space.*

