# OpenReview forum: "Exploring Reprensentation Invariance in Finetuning"
_ICLR.cc/2026/Conference — ICLR 2026 Conference Withdrawn Submission_

### Official Review · Reviewer_Y1qs · 2025-10-27

**Soundness:** 1
**Presentation:** 2
**Contribution:** 1
**Rating:** 2
**Confidence:** 5

**Summary:**

The paper proposes RIFT (Representation Invariance Fine-Tuning), a regularization strategy for fine-tuning vision foundation models on low-resource, cross-domain tasks (e.g., medical imaging). Motivated by the Platonic Representation Hypothesis, the authors argue that enforcing last-layer representation similarity between a pretrained and a finetuned model up to orthogonal transformations can preserve the generalizability of pretrained features. Concretely, they replace explicit CKA-based similarity constraints with an orthogonality-based covariance-matching objective applied to the final feature embeddings, and show that RIFT can be combined with PEFT/adapter methods. Empirically, RIFT improves representation similarity and provides competitive (sometimes slightly improved) task performance.

**Strengths:**

- **Timely problem & clear motivation:** The paper addresses representational drift during cross-domain, low-resource fine-tuning which is an issue of clear interest to the ICLR community, particularly for medical imaging and other distribution-shift domains.
- **Conceptual clarity & simplicity:** Formalizing a similarity-invariant parameter subspace via last-layer embeddings is a simple and understandable design choice. Equation (5) is particularly clean and intuitive.
- **Practical integration:** RIFT is “plug-and-play” and compatible with standard fine-tuning/PEFT methods without architectural surgery, which increases practical relevance.
- **Empirical breadth within the medical domain:** Experiments span multiple medical datasets and baselines (FULL, LINEAR, LoRA, AdaptFormer, VPT).
- **Useful analyses & visuals:**
    - PCA plots illustrate distributional overlap between pretrained and finetuned representations and support the claim that RIFT preserves semantic structure.
    - Sharpness-based analysis provides a quantitative angle on generalization/robustness.
- **Appendix materials:** Extra ablations and efficiency measurements help understand trade-offs (e.g., λ, batch size, training time).

**Weaknesses:**

# Major Concerns

1. **Motivational gap between parameter initialization and representation invariance.**
    The introduction argues that fine-tuning “implicitly minimizes” distance in parameter space, but then pivots to enforcing representation similarity. It remains unclear why representation invariance (rather than parameter proximity) is necessary or even desirable for single-task, cross-domain fine-tuning. Citations offered in the introduction primarily support the utility of pretrained parameters as strong initializations, not the necessity of preserving representations. The core insight here feels under-justified for single-task adaptation.

2. **Single-task setting vs. claimed benefits.**
    Retaining pretrained representations is often most beneficial for continual or multi-task learning. In this paper’s single-task setup, performance gains are small or mixed. This undermines the claim that preserving last-layer similarity is needed for effective low-resource adaptation.

3. **Unclear constraint story and missing theory where it matters.**
    The paper suggests pretrained and finetuned representations fail to coexist due to “insufficient constraints,” but the exact constraints that matter are not crisply specified or theoretically supported. The protein folding analogy is interesting but not operationalized.

4. **Orthogonality only at the last layer: rationale vs. practice.**
    The method enforces invariance at the final layer to bypass violations in deep, nonlinear models. However, the evaluated backbones (ViT, DINOv2) are deep and nonlinear. The argument that multi-layer/nonlinear activations break invariance in general appears to undercut the method’s own applicability; why should last-layer covariance matching translate into meaningful invariance or stability across the earlier nonlinear blocks?

5. **Objective transition (Eq. 5 $\to$ Eq. 6) is underspecified.**
    Equation (6) effectively sets the hard constraint’s margin $\epsilon$ in (5) to zero via a Lagrangian relaxation, yet the paper does not justify this choice or explore hinge-style relaxations. This is the weakest theoretical step and likely affects empirical behavior.

6. **Computational motivation is unconvincing as stated.**
    The paper argues that explicit CKA is expensive, but the complexity comparisons (w.r.t. LoRA or other PEFT ranks) are not made apples-to-apples. Since $d$ is the model width and $n$ the batch size, the claimed savings vs. strong baselines are not convincingly demonstrated or benchmarked. The efficiency story needs clearer, quantitative comparisons against realistic alternatives.

7. **Ambiguity around mean/covariance alignment and BatchNorm analogy.**
    The paper moves from mean+covariance alignment to pure covariance ($\mu$ implicitly treated as $0$ in the main loss), but the justification is brief. Moreover, the distinction between RIFT’s running statistics and what BatchNorm already does is not clearly articulated. A concise algorithm/pseudocode would help.

8. **Section 4.3 toy examples feel misaligned.**
    - The autoencoder rotation example (Fig. 2) is not persuasive: decoding after a latent rotation producing a rotated image is expected; it does not substantiate last-layer feature orthogonal invariance as used in RIFT.

    - The 6-layer linear network in Fig. 3 is effectively a single linear map; its relevance to deep nonlinear ViTs is weak. The section’s conclusion (“invariance breaks with nonlinearity”) appears to conflict with RIFT’s goal and evaluation setting.

9. **Baselines & reporting clarity.**

    - “RSC” appears in Tables 1 and 2 but is not clearly defined in the main text or appendix (implementation details, $\lambda$-tuning, and whether results are reproduced or copied are unclear).

    - Several implementation choices (e.g., LoRA rank) are unspecified; it’s thus hard to interpret the fairness of comparisons.

    - APTOS2019 in Table 2: FULL (83.61%) outperforms FULL+RIFT* (83.06%), yet the text claims otherwise. This needs correction.

10. **Core hypothesis not strongly supported by the numbers.**
    Across Tables 1-3, higher similarity does not consistently correlate with higher task performance; in several cases, the strongest performance has lower or comparable similarity to RIFT. This challenges the central claim that preserving representation similarity is essential for low-resource fine-tuning.

## Minor Concerns

1. **Sharpness results (Table 4):** It is surprising that LINEAR does not consistently yield the lowest sharpness; additional detail on baseline implementations would build confidence.

2. **Ablations (Table 5/6):**

    - Datasets used for ablations are not stated; please specify per table.

    - If a learnable $Q$ improves results, justify why it is not part of the main method settings.

    - Table 6 shows increasing similarity sometimes reduces accuracy. Please discuss this trade-off explicitly.

3. **Figure/Table formatting:**

    - Bold the tied best entries (e.g., VPT and VPT+RIFT* on APTOS2019; LoRA and LoRA+RIFT* on APTOS2019). This will make clear that **gains are often within noise.**

4. **Appendix A3 (linear networks):** A minimal Linear-ReLU-Linear counterexample would better match the nonlinear ViT setting than purely linear stacks.

5. **Terminology/consistency:** Clarify whether Table A4 truly reflects multi-task training; the main text suggests single-task fine-tuning, and the MTL setup is not described.

**Questions:**

1. **On disabling adapters:**
    If the concern is losing pretrained representations, why not freeze the backbone and train separate LoRA/adapters per task (for continual learning/MTL)? How does RIFT compare to simple “disable adapter” or “frozen backbone” strategies regarding generalization and performance on in-domain and cross-domain evaluations?

2. **From parameter initialization to representation invariance:**
    Please clarify the logical step from “initializing with pretrained parameters” to “enforcing high representation similarity” when the downstream domain differs from pretraining. Under what conditions is last-layer similarity sufficient (or necessary) for transfer?

3. **Baselines for representation preservation:**
    Methods that preserve basis/singular spectra (e.g., DiTASK (Mantri et al., CVPR 2025), SVFT (Lingam et al., NeurIPS 2024), SVF (Sun et al., NeurIPS 2022)) appear directly relevant. Why were these omitted, and how would RIFT compare?

    >Mantri et al., DiTASK: Multi-Task Fine-Tuning with Diffeomorphic Transformations., CVPR 2025.
    >Lingal et al., SVFT: Parameter-Efficient Fine-Tuning with Singular Vectors., NeurIPS 2024.
    >Sun et al., Singular Value Fine-tuning: Few-shot Segmentation requires Few-parameters Fine-tuning., NeurIPS 2022.

4. **Similarity metric choice:**
    Given concerns about CKA’s reliability, why not evaluate with Latent Functional Maps (LFM; Fumero et al., NeurIPS 2024) in addition to CKA? Including LFM would strengthen the representational analysis.

    >Fumero et al., Latent Functional Maps: a spectral framework for representation alignment., NeurIPS 2024.

5. **Constraint formulation (ε and hinge):**
    Why is $\epsilon$ implicitly set to 0 in the Lagrangian relaxation (Eq. 5 $\to$ Eq. 6)? Have you tried a hinge-style penalty or margin-based formulation?

6. **Implementation specifics for Eqs. (10)-(12):**

    - Please provide pseudocode that shows how batch-wise covariances are computed and accumulated, how $Q$ is optimized, and how storage/compute are handled without double forward passes.

    - Clarify how this differs from BatchNorm’s running statistics.

    - If the pretrained covariance is required, is it computed on-the-fly per batch or precomputed? What are the memory implications?

7. **Section 4.3 / Figure 2–3 clarity:**

    - In Fig. 2, why is the decoder’s output used as the “feature”? In autoencoders, the latent is the standard representation. Please clarify the intent and relevance to last-layer feature invariance.

    - Fig. 3 uses a 6-layer linear network; can you replace this with Linear-ReLU-Linear (or a small MLP) to reflect the nonlinear case you argue about throughout?

8. **Experimental settings & fairness:**

    - Precisely define “RSC,” list all hyperparameters (including LoRA rank), and clarify whether baselines are reproduced or taken from prior work.

    - Report #learnable parameters for all methods and include ablations with matched parameter budgets across methods.

    - For Table 2, please also compare ViT-L and DINOv2-L to keep backbones within the same family.

9. **Why modest gains if similarity matters?**
    Many improvements are within a few tenths to ~1–2% absolute. How do you reconcile the small gains with the hypothesis that preserving representation similarity is essential? Under what regimes (data size, domain gap, $\lambda$) does RIFT most help?

10. **Ablations coverage & dataset specification:**
    Please rerun/expand ablations in Section 5.4 on MedFM-Endo, MedFM-Colon, and APTOS2019, and explicitly state the datasets used in each table/figure.

---

### Official Review · Reviewer_88DA · 2025-10-28

**Soundness:** 4
**Presentation:** 4
**Contribution:** 4
**Rating:** 6
**Confidence:** 4

**Summary:**

The paper proposes Representation Invariance Fine-Tuning (RIFT), a regularization approach compatible with standard fine-tuning methods. The idea is to maximize therepresentation similarity between pretrained and finetuned models by leveraging the orthogonal invariance property of manifolds. Rather than relying on computationally heavy pairwise similarity measures (as in RSC), RIFT constrains the finetuned representation to remain on an orthogonal manifold of the pretrained one by introducing a covariance-based regularization term in the loss function.
Experiments on several medical imaging datasets demonstrate that RIFT effectively enhances representation similarity (CKA), thus preserving the generalization ability of the pretrained model while keeping downstream task performance competitive. Additional evaluations on larger backbones (ViT-Large, DINOv2) and zero-shot generalization tasks further confirm the method’s robustness and its capacity to retain transferable representations.

**Strengths:**

Clarity: the paper is well written, well structured, and the motivation is clearly explained.

Elegance of the method: the proposed approach is conceptually elegant and straightforward to understand. Its theoretical foundation is solid, being well supported by property 3.1 and Theorem A3.5. In essence, by including a regularization term in the loss function, the finetuned representation is encouraged to reside on an orthogonal manifold of the pretrained representation, thereby preserving semantic and structural similarity.

Efficiency: unlike other methods, RIFT does not require a sample-wise forward pass through the pretrained model. It performs distributional alignment by matching the covariance (and optionally mean) of mini-batch features, reducing the computational complexity from (O(nd^2)) to (O(d^2)). This makes it both elegant and practical for large-scale fine-tuning.

Comprehensive experiments and ablations: the authors support their claims with extensive quantitative and qualitative evaluations, including ablations on the orthogonal matrix (Q), batch size, and the regularization weight λ. The decision to apply orthogonal constraints only on final embeddings (and not intermediate layers) is well argued and experimentally validated (Figures 2 and 3). The sharpness-based generalization study (Table 4) and the PCA visualizations (Figure 4) provide convincing evidence of improved representational alignment.

Compatibility: the method is plug-and-play and works with both full finetuning and PEFT approaches such as LoRA, AdaptFormer, and VPT. The consistent improvement in CKA similarity across all setups supports the claimed generality of the method.

**Weaknesses:**

Modest downstream gains: while RIFT consistently enhances representation similarity (CKA), the corresponding improvement in accuracy or mAP over standard fine-tuning or RSC is sometimes inconsistent. The practical impact in terms of performance gains may therefore be limited, despite its conceptual novelty.

Domain limitation: all experiments are conducted in the vision domain primarily on medical image classification tasks. The authors acknowledge this in the limitations, but evaluations on other modalities (e.g., language, multimodal) and domains would have significantly strengthened the generality and impact of the work.

Minor inconsistencies: There is a minor editorial issue, specifically the repetition of a sentence on line 327.

**Questions:**

1) While Theorem A3.5 provides a theoretical justification for covariance-level alignment, how robust is this assumption in practice? Could there be cases where covariance similarity fails to fully preserve the semantic structure of representations?
2. Could RIFT be extended to multimodal or cross-domain fine-tuning scenarios, where pretrained and finetuned representations come from different modalities?

---

### Official Review · Reviewer_1Pcq · 2025-10-31

**Soundness:** 2
**Presentation:** 2
**Contribution:** 1
**Rating:** 2
**Confidence:** 4

**Summary:**

The paper studies how to effectively finetune models without forgetting generalist pretrained representations.The authors name their technique, Representation Invariance FineTuning (RIFT), a regularization that maximizes the representation similarity between pretrained and finetuned models by leveraging orthogonal invariance of manifolds in a computationally efficient way.

**Strengths:**

1. The paper is well written and easy to understand
2. The authors conduct many experiments across different datasets, and zero-shot evaluation, etc.

**Weaknesses:**

I think the paper fails to tangibly establish the benefits of their proposed RIFT loss, as the task accuracy in most experiments remains nearly the same compared to full fine-tuning. While similarity improves, since it's now explicitly optimised to do so in their configuration, it's not entirely clear to me why that is tangibly better. One way to investigate this could have been a more exhaustive performance evaluation, including accuracy and other metrics on large-scale pretraining tasks with the finetuned model.

Even with such evaluations, I still struggle to fully appreciate the gravity of the problem. With LoRA adapters, for instance, one can simply swap them out to retain generalised pretrained representations, or even compose different LoRA adapters, and we would not necessarily suffer from forgetting.

Overall, I think the authors need a clearer experimental design to convincingly demonstrate the benefits of their method, beyond improvements in the specific metric they're optimizing for.

**Questions:**

See Weaknesses

---

### Official Review · Reviewer_oVCo · 2025-10-31

**Soundness:** 2
**Presentation:** 3
**Contribution:** 2
**Rating:** 2
**Confidence:** 4

**Summary:**

The paper addresses the problem of representation degradation and loss of generalizability in foundation models during fine-tuning. It introduces Representation Invariance Fine-Tuning (RIFT), a regularization approach that preserves pretrained representations by enforcing orthogonal invariance on the representation space. Extensive experiments on image classification tasks demonstrate that RIFT is compatible with mainstream fine-tuning strategies, achieving competitive or improved performance.

**Strengths:**

1. The paper explores an interesting direction by examining fine-tuning through the lens of representation invariance in foundation models, which offers useful insights into understanding fine-tuning dynamics.
2. The proposed RIFT loss is concise and appears easily integrable into modern foundation models with minimal engineering overhead.
3. The overall organization and writing is clear, and the visualization in Fig. 4 effectively demonstrates that the RIFT loss helps mitigate distribution shift.

**Weaknesses:**

Overall, I agree with the central motivation of preserving representational advantages during fine-tuning; however, after carefully reading the paper, several concerns arise.
1. The experiments are limited to a single-task transfer scenario. In this setting, models are not reused for further transfer, but only adapted to one downstream task. Thus, the motivation for enforcing representation invariance becomes unclear, since representation changes may not be detrimental if secondary transfer is not required. A stronger justification could be provided through settings such as multi-task learning or continual learning, where the value of representation invariance would be more evident.
2. The derivation from the Similarity-Invariant Parameter Subspace (based on CKA invariance) to the final RIFT loss involves substantial relaxation. It remains uncertain whether the RIFT loss enforces sufficient invariance constraints. In addition, the paper should explicitly discuss and compare RIFT with existing fine-tuning approaches that impose regularization in the parameter space [1] or in the feature space [2] and orthogonal optimization method such as [3]. Such comparisons should cover both downstream performance and representation similarity to clarify the advantages and limitations of the proposed method.
3. The trade-off between feature invariance and downstream performance is critical in practice. Results in Table 1 suggest that RIFT is highly sensitive to hyper-parameter choices, and a more principled approach beyond grid search (Table A2) or a more robust loss terms is needed to ensure consistent performance.
[1] Explicit inductive bias for transfer learning with convolutional networks. In ICML, 2018.
[2] Delta: deep learning transfer using feature map with attention for convolutional networks. In ICLR, 2018.
[3] Controlling text-to-image diffusion by orthogonal finetuning. In NeurIPS, 2023.

**Questions:**

No additional questions; please see weaknesses.

---

### Note · Authors · 2025-12-01

I have read and agree with the venue's withdrawal policy on behalf of myself and my co-authors.